

# 3D model retrieval based on interactive attention CNN and multiple features

Xue-Yao Gao, Wen-Hui Jia and Chun-Xiang Zhang

School of Computer Science and Technology, Harbin University of Science and Technology, Harbin, China

## ABSTRACT

3D (three-dimensional) models are widely applied in our daily life, such as mechanical manufacture, games, biochemistry, art, virtual reality, and *etc*. With the exponential growth of 3D models on web and in model library, there is an increasing need to retrieve the desired model accurately according to freehand sketch. Researchers are focusing on applying machine learning technology to 3D model retrieval. In this article, we combine semantic feature, shape distribution features and gist feature to retrieve 3D model based on interactive attention convolutional neural networks (CNN). The purpose is to improve the accuracy of 3D model retrieval. Firstly, 2D (two-dimensional) views are extracted from 3D model at six different angles and converted into line drawings. Secondly, interactive attention module is embedded into CNN to extract semantic features, which adds data interaction between two CNN layers. Interactive attention CNN extracts effective features from 2D views. Gist algorithm and 2D shape distribution (SD) algorithm are used to extract global features. Thirdly, Euclidean distance is adopted to calculate the similarity of semantic feature, the similarity of gist feature and the similarity of shape distribution feature between sketch and 2D view. Then, the weighted sum of three similarities is used to compute the similarity between sketch and 2D view for retrieving 3D model. It solves the problem that low accuracy of 3D model retrieval is caused by the poor extraction of semantic features. Nearest neighbor (NN), first tier (FT), second tier (ST), F-measure (E(F)), and discounted cumulated gain (DCG) are used to evaluate the performance of 3D model retrieval. Experiments are conducted on ModelNet40 and results show that the proposed method is better than others. The proposed method is feasible in 3D model retrieval.

## INTRODUCTION

3D modeling is coming into our daily life. It is convenient to obtain sketch, and sketch can describe a 3D model's shape. It is in line with people's habits to retrieve 3D models based on a sketch. However, a sketch contains little shape information, and there is a big gap between sketch and 3D model. It causes low accuracy of 3D model retrieval.

3D model retrieval includes text-based retrieval algorithm, content-based retrieval algorithm and sketch-based retrieval algorithm. In text-based retrieval algorithm, 3D model is manually described according to its shape and topological structure. According to user's description, 3D model which meets design intent is retrieved. In content-based

Corresponding author
Chun-Xiang Zhang,
z6c6x666@163.com

algorithm, 3D model's information is used for 3D model retrieval. *Tangelder & Veltkamp (2008)* surveyed the literature on methods for content-based 3D model retrieval, taking into account the applicability to surface models as well as to volume models. It can be divided into three methods including view-based algorithm, voxel-based algorithm and point cloud-based algorithm. In view-based algorithm, 3D model is converted into a series of 2D views. Then, features of 2D view are calculated and compared for 3D model retrieval. In voxel-based algorithm, 3D model is expressed as the distribution of 3D voxel grid. Features of voxel data are extracted for 3D model retrieval. In point cloud-based algorithm, feature descriptors are directly extracted from original point cloud to retrieve 3D models. In sketch-based algorithm, 3D models are usually converted into 2D images from which features are extracted to realize 3D model retrieval. Main contributions of this article are summarized as follows:

(1) Interactive attention module is embedded into CNN, in which attention weights of two adjacent convolutional layers are merged.

(2) D1, D2 and D3 are used to describe shape of sketch and 2D view, and gist feature is used to describe spatial structure of sketch and 2D view.

(3) Semantic feature, shape distribution features and gist feature are combined adaptively to find 3D model similar to sketch.

## Literature review

In text-based algorithm, it is necessary to describe shape and topological structure of 3D model manually. *Tian et al. (2012)* proposed a graph-based method to expand labelled data. Multi-label lazy learning approach was given based on its k nearest neighbors, and maximum *a posteriori* (MAP) principle was utilized to determine its category. *Jandial et al. (2022)* gave a novel semantic attention composition framework for text-conditioned image retrieval including semantic feature attention and semantic feature modification. However, this method can only retrieve 3D model by tags, and cannot retrieve 3D model based on its content.

In view-based algorithm, 3D model is converted into a set of images through stereographic projection. *Bu et al. (2014)* applied deep belief network to extract shape features to retrieve 3D model. *Lu et al. (2015)* used view-model relevance for 3D model retrieval. *Gao et al. (2022)* proposed a novel and effective multi-level view associative convolution network to realize view-based 3D model retrieval. Although this method has achieved good effect, it is difficult to obtain all information of 3D model from 2D views.

In voxel-based algorithm, 3D voxel matrix is used to denote 3D model. *Osada, Furuya & Ohbuchi (2008)* described a shape-based retrieval method that extracted local and multi-scale features from voxel representation of 3D model. *Wang et al. (2019)* proposed a voxel-based convolutional neural network, which used normal vector of model surface and voxel matrix of 3D model. *Zhang et al. (2021)* retrieved 3D model by combining 3D voxel model with octree structure. However, voxel resolution affects the performance of 3D model retrieval. High resolution increases the retrieval complexity, and *vice versa*.

In point cloud-based algorithm, feature descriptors are extracted from point cloud. *Lu, Zhang & Liu (2019)* used L1-middle skeleton to represent spatial structures of 3D point cloud. *Uy & Lee (2018)* used end-to-end deep neural network to train and infer global descriptors from 3D point cloud. *Li et al. (2020)* proposed a multi-part attention network for 3D model retrieval based on point cloud. But, point cloud is irregular and disordered. It is difficult for convolution operators to take advantage of spatial local correlation in data, which affects the learning ability of neural networks to a certain extent.

In sketch-based algorithm, 3D model is projected into a set of 2D views. *Zhang et al. (2017)* used PCA-DAISY descriptor and fisher coding algorithm to retrieve 3D model according to 2D sketch. *Nie, Wang & Lu (2018)* proposed Multi-Scale and Multi-Channel CNN to extract features for 3D model retrieval. *Siddiqua & Fan (2019)* gave asymmetric supervised deep autoencoder to retrieve 3D shapes based on depth images. *Kuang, Yu & Zhu (2019)* integrated local descriptor with deep pointwise convolutional network to extract 1D features for shape recognition and retrieval. *Chen, Wang & Liu (2020)* used loop CNN to construct cross-domain mapping between 2D sketch and 3D shape, which was applied to 3D shape retrieval. *Yang et al. (2022)* proposed a novel sequential learning framework to learn 3D model's representation and 2D sketch's representation separately and sequentially. *Gao et al. (2020)* gave a novel way for retrieving 3D model by means of sketching and building the retrieval framework based on deep learning. *Jiao et al. (2020)* proposed the cross-domain correspondence method for the sketch-based retrieval based on manifold ranking. *Bai et al. (2019)* gave an end-to-end retrieval framework of retrieving 3D model according to sketch based on joint feature mapping. *Tan, Fan & Guo (2018)* proposed an improved manifold ranking method, where all categories between arbitrary model pairs were taken into account. *Lei et al. (2018)* gave a new sketch-based 3D model retrieval approach which used two kinds of semantic attributes, including pre-defined attributes and latent attributes. *Liu et al. (2020)* introduced a method for generating line drawings from 3D models. Geometric and view-based reasoning were combined with the help of neural module to create a line drawing. *Qin et al. (2017)* designed a fine granularity semantic descriptor to represent 3D models, adopted heuristic rules to recognize 3D features from 2D sketch, and built the correspondences between 3D feature and 2D loops. *Li et al. (2014)* performed comprehensive comparison of 15 best retrieval methods by completing the evaluation of each method on both benchmarks.

The dimension of 3D model is 3, and the dimension of sketch is 2. It is key to express 3D model and sketch correctly for the sketch-based retrieval. In order to compute the similarity between sketch and 3D model, we should express them in the same dimension space. So, we project 3D model into two dimension space from different angles and use multiple 2D projections to describe 3D model. We extract feature vectors from sketch and 2D projections of 3D model. Then, the similarity between sketch and 2D projections is computed for retrieving 3D model similar to sketch.

**Table 1 The number of 2D views and sketches in each category.**

|  | 2D view | Sketch |
|---|---|---|
| Airplane | 144 | 140 |
| Car | 144 | 140 |
| Chair | 144 | 140 |
| Cup | 144 | 138 |
| Dresser | 126 | 137 |
| Lamp | 144 | 137 |
| Person | 174 | 88 |
| Plant | 114 | 152 |
| Table | 144 | 141 |

## MATERIALS AND METHODS

### SHREC13 and ModelNet40

ModelNet40 dataset is used in this experiment. SHREC13 is used as the sketch library. In order to verify the proposed method, nine categories are selected that are the same in SHREC13 and ModelNet40. These nine categories are also contained in SHREC14. They are respectively airplane, car, chair, cup, dresser, lamp, person, plant and table. Models are randomly selected from training set of each class in SHREC13 and ModelNet40. At the same time, 2D views are extracted from these models.

In retrieval stage, target objects are selected from SHREC13. Test data come from ModelNet40. In order to enhance the improved CNN, training data come from SHREC13 and ModelNet40. The purpose is to make the improved CNN consider shape distribution of sketches in SHREC13 and 3D models in ModelNet40. Six 2D views are extracted from 3D model. 2491 sketches and 2D views are used as training data. The number of 2D views and sketches in each category are shown in Table 1. A total of 540 2D views of 3D model are used as test data. In retrieval stage, nine sketches from SHREC13 are target objects and used to evaluate the retrieval performance.

### Methods

#### Retrieve 3D model based on sketch

When 3D model is retrieved based on sketch from model library, it is necessary to extract features from sketch and 3D models. The dimension of sketch is 2, and the dimension of 3D model is 3. The fixed projection algorithm is used in this article. 3D model is fixed to the center of virtual sphere, where virtual camera is placed above it. The model is rotated one circle to render a set of 2D views at 60° every step. In order to reduce texture difference between sketch and 2D views, edge detection algorithm is used to extract contours from sketch and 2D views.

CNN is adopted to extract semantic feature of view and sketch. Meanwhile, gist feature and shape distribution features of view and sketch are also used as global features. Euclid distance is used to calculate the similarity between sketch and 3D model's view. These three features are combined to compute the similarity between sketch and 3D model.

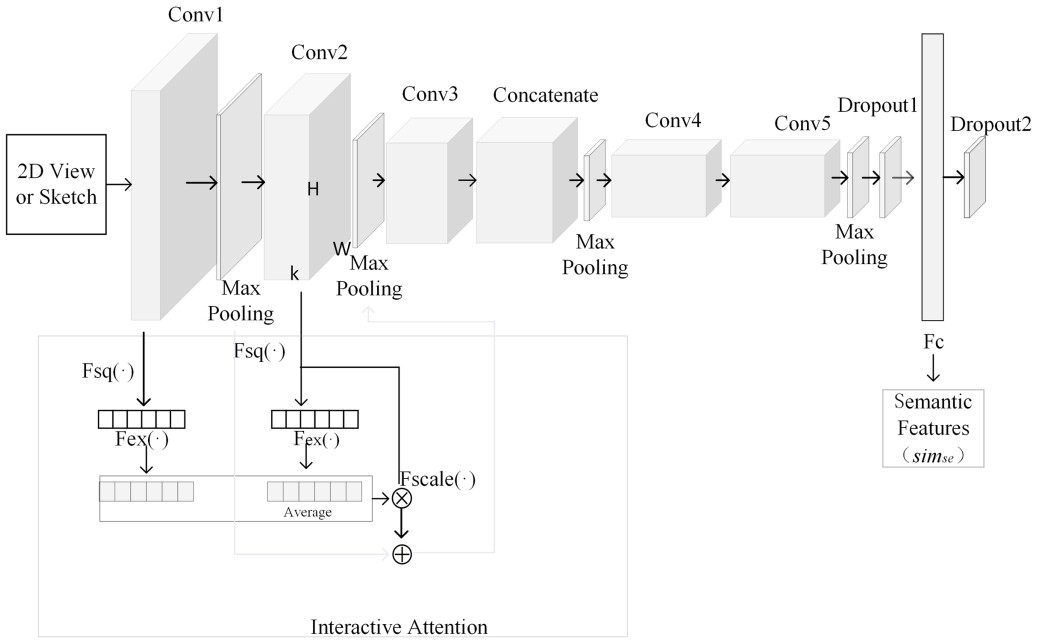

**Figure 1 The process of extracting semantic features based on CNN.**

Feature of good discriminative ability is assigned with high weight. Otherwise, it is assigned with low weight. The weighted sum of these three features is used to compute the similarity between sketch and 3D model. Then, similar model is retrieved from 3D model library.

### Feature extraction of 3D model and sketch

1. Semantic feature

In this article, CNN is adopted to extract semantic features from 3D model's views and sketch. The process of extracting semantic features based on CNN is shown in Fig. 1. The above CNN is composed of five convolutional layers, four pooling layers, an attention layer, a fusion layer, a dropout layer, a fully connected layer and an output layer. In order to reduce the influence of image texture on 3D model retrieval, 2D view is rendered as contour. Input layer of convolutional neural network can accept sketch and contour of 2D views. Since fully connected layer requires a fixed input dimension, the size of sketch should be the same with that of 2D view. Output of convolutional layer is shown in Formula (1).

$$a^l = f\left(a^{l-1} * K^l + b^l\right) \tag{1}$$

where, $l$ is the number of layers, $K^l$ is convolution kernel matrix, $b^l$ denotes bias, as asterisk (*) represents convolution operation. Here, $f$ is Relu function. Output of pooling layer is shown in Formula (2).

$$a^l = max\left(a^{l-1}\right) \tag{2}$$

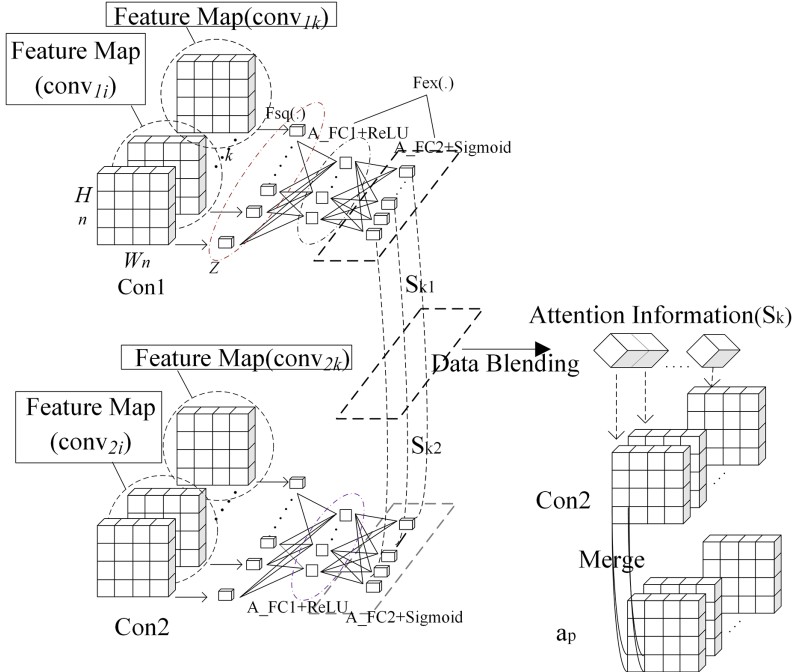

**Figure 2  Interaction attention module.**

Output of fully connected layer is shown in Formula (3).

$$a^l = f(W^l * a^{l-1} + b^l) \tag{3}$$

where, $W^l$ represents weight matrix in fully connected layer, $b^l$ denotes bias. Interactive attention module is introduced into the framework as shown in Fig. 1. Output from Conv1 is processed by attention to get its attention weight. Output from Conv2 is processed by attention to get its attention weight. These two weights are averaged. The averaged weight is fused with output from Conv2. The result is summed with the max pooling of output from Conv1, which is output to the second max pooling layer. Output from Conv2 is processed with attention weights of output from Conv1 and Conv2. The purpose is to fuse more convolutional layers to improve the quality of semantic features. Interaction attention module is shown in Fig. 2. In order to calculate information quantity in each channel of the first and second convolutional layer, pooling operation with global receptive field is used to compress spatial information. Then, feature map $conv_{nk}$ is converted into a real number as shown in Formula (4). All real numbers are used to construct vector $Z$.

$$Z_k = F_{sq}(conv_{nk}) = \frac{1}{H_n * W_n} \sum_{i=1}^{H_n} \sum_{j=1}^{W_n} conv_{nk}(i,j) \tag{4}$$

where, $conv_{nk}$ represents the $k$th feature map in output of the $n$th convolutional layer, and its size is $W_n * H_n$. In order to tune attention weight of each channel adaptively, two fully connected layers are connected after global pooling layer. The first one $A\_C_1$ has K/r neurons. Here, r is scaling ratio. The second one $A\_FC_2$ has K neurons. Attention weight $S_{kn}$ is computed as shown in Formula (5).

$$S_{kn} = F_{ex}(Z, W) = \sigma(g(Z, W)) = \sigma(W_2\delta(W_1Z)) \tag{5}$$

where, $\delta$ is Relu function and $\sigma$ is sigmoid function. $S_{kn}$ is weight vector of convolutional layer $Conn$ and its dimension is $k$. $W_1$, $W_2$ are respectively weights of $A\_FC_1$ and $A\_FC_2$. The algorithm of computing interaction attention is shown as follows:

Input: feature map $con_{v1} = (conv_{11}, ..., conv_{1k})$ of the first convolutional layer $Con_1$, feature map $conv_2 = (conv_{21}, ..., conv_{2k})$ of the second convolutional layer $Con_2$, feature map $a_p$ of the first pooling layer.

Output: feature map $a_2$ of the second convolutional layer $Con_2$.

① Calculate vector $Z_k$ of $conv_{1i}$ ($i = 1, 2, ..., k$) in the first convolutional layer $Con_1$ according to Formula (4).
② Calculate attention weight $S_{k1}$ of convolutional layer $Con_1$ according to Formula (5).
③ Calculate vector $Z_k$ of $conv_{2i}$ ($i = 1, 2, ..., k$) in the second convolutional layer $Con_2$ according to Formula (4).
④ Calculate attention weight $S_{k2}$ of convolutional layer $Con_2$ according to Formula (5).
⑤ Fuse $S_{k1}$ and $S_{k2}$ according to Formula (6).

$$S_k = Average(S_{k1}, S_{k2}) \tag{6}$$

where, $Average$ (.) is average function.

⑥ Attention weight $S_k$ is fused with $conv_2$ and $a_p$ as shown in Formula (7).

$$a_2 = conv_2 \otimes s_k + a_p \tag{7}$$

2. Gist feature
Gist feature is global feature that contains similar shape and spatial structure. The process of extracting gist feature is shown as follows:

(1) Divide sketch or 2D view with size m * n into 4 * 4 blocks. The size of each block is a * b, where $a = m/4$, $b = n/4$.

(2) Each block is processed by 32 Gabor filters with four scales and eight directions. Processed features are combined to obtain gist feature as shown in Formula (8).

$$G(x, y) = \underset{32}{cat}\left(I(x, y) * g_{ij}(x, y)\right) \tag{8}$$

where, $i = 4$, $j = 8$. $G(x, y)$ is gist feature of 32 Gabor filters, and $cat()$ represents concatenation operation. Here, $x$ and $y$ are positions of pixel, $I(x, y)$ denotes the block. At the same time, $g_{ij}(x, y)$ is filter with the $i$th scale and the $j$th direction. Here, * represents convolution operation.

(3) Feature $G(x, y)$ is averaged after each filtering. All average block features are combined into a row vector to obtain final gist block feature.

(4) Features of 16 blocks are combined to form gist feature of 2D view.

3. Shape distribution features

Osada proposes shape distribution for 3D model's representation which is used to retrieve and classify 3D models (*Osada, Funkhouser & Chazelle, 2001*). We modify 3D model's shape descriptors and apply them to 2D views. Points on the boundary of sketch or 2D view are randomly sampled equidistantly, which are collected into *Points* = $\{(x_1, y_1), ..., (x_i, y_i), ..., (x_n, y_n)\}$. Here, $(x_i, y_i)$ is point coordinates. Extract points from *Points* and collect them into $PD1 = \{ai_1, ..., ai_k, ..., ai_N\}$. The set of D1 shape distribution feature is $\{D1\_v_1, ..., D1\_v_i, ..., D1\_v_{Bins}\}$. Here, $D1\_v_i$ is statistics in interval (*BinSize* * $(i-1)$, *BinSize* * $i$), *Bins* represents the number of intervals, and *BinSize* is interval length. $D1\_v_i$ is computed as shown in Formula (9).

$$D1\_v_i = |\{P|dist(P, O) \in (BinSize * (i-1), BinSize * i), P \in PD1\}| \tag{9}$$

where, $BinSize = max(\{dist(P, O)|P \in PD1\})/N$, $dist()$ is Euclidean distance between two points. $O$ is the centroid of sketch or 2D view. Extract point pairs from *Points* and collect them into $PD2 = \{(ai_1, bi_1), (ai_2, bi_2), ..., (ai_N, bi_N)\}$. The set of D2 shape distribution feature is $\{D2\_v_1, ..., D2\_v_i, ..., D2\_v_{Bins}\}$. Here, $D2\_v_i$ represents statistics in interval (*BinSize* * $(i-1)$, *BinSize* * $i$). $D2\_v_i$ is calculated as shown in Formula (10).

$$D2\_v_i = |\{P|dist(P) \in (BinSize * (i-1), BinSize * i), P \in PD2\}| \tag{10}$$

where, $BinSize = max(\{dist(P)| P \in PD2\})/N$. Extract point triples from *Points* and collect them into $PD3 = \{(ai_1, bi_1, ci_1), (ai_2, bi_2, ci_2), ..., (ai_n, bi_n, ci_n)\}$. The set of D3 shape distribution feature is $\{D3\_v_1, ..., D3\_v_i, ..., D3\_v_{Bins}\}$. Here, $D3\_v_i$ represents statistics in interval (*BinSize* * $(i-1)$, *BinSize* * $i$). $D3\_v_i$ is computed as shown in Formula (11).

$$D3\_v_i = |\{P|herson(P) \in (BinSize * (i-1), BinSize * i), P \in PD3\}| \tag{11}$$

where, $BinSize = max(\{\sqrt{herson(P)}|(P) \in PD3\})/N$. Here, $herson()$ represents Helen formula, which is used to compute area of triangle $P = (P_1, P_2, P_3)$ as shown in Formulas (12) and (13).

$$herson(P) = herson(P_1, P_2, P_3) = \sqrt{s(s-a)(s-b)(s-c)} \tag{12}$$

$$s = \frac{1}{2} * (a + b + c) \tag{13}$$

where, $a = dist(P_1, P_2)$, $b = dist(P_1, P_3)$, $c = dist(P_2, P_3)$. $D1\_v_i$, $D2\_v_i$ and $D3\_v_i$ are connected to form shape distribution feature, $i = 1, 2, ..., Bins$.

### Similarity fusion and model retrieval

The distance between features is used to measure the similarity between sketch and 2D view. Feature of sketch is denoted as $X_s = (x_{s1}, x_{s2}, ..., x_{sn})$. 2D view's feature is represented as $X_v = (x_{v1}, x_{v2}, ..., x_{vn})$. Euclidean distance between $X_s$ and $X_v$ is computed as shown in Formula (14).

$$D_{Euc}(X_s, X_v) = |X_s, X_v| \tag{14}$$

Algorithm of calculating the similarity between sketch and 2D view is shown as follows:
Input: sketch $A$ and 2D view $v$
Output: the similarity between $A$ and $v$

**Table 2 Accuracy of CNN under different convolutional layer number and pooling layer number.**

| Pooling layer number<br>Convolutional layer number | 3 | 4 | 5 | 6 |
|---|---|---|---|---|
| 3 | 0.936 | 0.923 | 0.913 | 0.765 |
| 4 | 0.949 | 0.956 | 0.950 | 0.923 |
| 5 | 0.953 | 0.964 | 0.960 | 0.958 |
| 6 | 0.923 | 0.944 | 0.962 | 0.954 |

1. Use the improved CNN to extract $X_{s1}$ from $A$ and normalize $X_{s1}$.
2. Use the improved CNN to extract $X_{v1}$ from $v$ and normalize $X_{v1}$.
3. Calculate $sim_{se}(X_{s1}, X_{v1}) = 1/D_{Euc}(X_{s1}, X_{v1})$ by Formula (14).
4. Extract gist feature from $A$ to get $X_{s2}$ and normalize $X_{s2}$.
5. Extract gist feature from $v$ to get $X_{v2}$ and normalize $X_{v2}$.
6. Calculate $sim_{gi}(X_{s2}, X_{v2}) = 1/D_{Euc}(X_{s2}, X_{v2})$ by Formula (14).
7. Extract shape distribution feature from $A$ to get $X_{s3}$ and normalize $X_{s3}$.
8. Extract shape distribution feature from $v$ to get $X_{v3}$ and normalize $X_{v3}$.
9. Calculate $sim_{sh}(X_{s3}, X_{v3}) = 1/D_{Euc}(X_{s3}, X_{v3})$ by Formula (14).
10. Fuse similarities by Formula (15).

$$Sim(A, v) = w_1 \times sim_{se}(X_{s1}, X_{v1}) + w_2 \times sim_{gi}(X_{s2}, X_{v2}) + w_3 \times sim_{sh}(X_{s3}, X_{v3}) \qquad (15)$$

where, $0 < w_i < 1$, $i = 1, 2, 3$, $w_1 + w_2 + w_3 = 1$.

## EXPERIMENTAL RESULTS AND DISCUSSION

The proposed method can extract semantic feature, gist feature and shape distribution features from 2D views. If we get 2D views of 3D model, we can use the proposed method to retrieve 3D model according to sketch. This article uses NN, FT, ST, E(F), DCG to evaluate the performance of 3D model retrieval.

The performance of CNN is determined by convolutional layer number and pooling layer number. Convolutional layer number and pooling layer number are respectively set to 3, 4, 5, 6. When convolutional layer number is 3 and pooling layer number is 3, each convolutional layer is followed by a pooling layer. When convolutional layer number is 4 and pooling layer number is 3, each of the first three convolutional layers is followed by a pooling layer. CNN is optimized and tested as shown in Table 2.

From Table 2, we can find that CNN achieves the best and accuracy is 0.964 when convolutional layer number is 5 and pooling layer number is 4. So, the proposed network contains five convolutional layers and four pooling layers.

Here, accuracy of test data is used to evaluate the quality of feature extraction. At the same time, accuracy of test data is also adopted to determine position and way of adding channel attention into CNN.

AT-CNN1 represents classifier where AT (attention module) is only added into the first convolutional layer. AT-CNN2 is classifier where attention module is added between the first convolutional layer and the second one. AT-ALL-CNN denotes classifier where

**Table 3 Classification accuracy under different dropout coefficients and different ways of adding channel attention into CNN.**

| Method | AT-CNN1 | AT-CNN2 | AT-ALL-CNN | ICA-12 | ICA-23 |
|---|---|---|---|---|---|
| Dropout = 0.1 | 0.923 | 0.916 | 0.906 | 0.958 | 0.944 |
| Dropout = 0.2 | 0.936 | 0.927 | 0.922 | 0.958 | 0.939 |
| Dropout = 0.3 | 0.941 | 0.921 | 0.923 | 0.963 | 0.954 |
| Dropout = 0.4 | 0.941 | 0.923 | 0.933 | 0.969 | 0.967 |
| Dropout = 0.5 | 0.938 | 0.919 | 0.919 | 0.933 | 0.944 |

attention module is added into all convolutional layers. ICA-12 represents classifier where interaction weight attention is added between the first convolutional layer and the second one. ICA-23 is classifier where interaction weight channel attention is added between the second convolutional layer and the third one. In order to prevent the overfitting problem, we add dropout into CNN and set different dropout coefficients. Table 3 shows accuracy under different dropout coefficients and different ways of adding channel attention into CNN.

From Table 3, it can be seen that when more attention modules are added into CNN, the lower is accuracy. Channel attention mechanism can focus on some channels. The reason may be that there are few lines in sketch and 2D views, and accuracy will be reduced if information of some channels is ignored. Interaction weight channel attention can reduce information loss to a certain extent and improve the performance of 3D model retrieval. This is because that channel attention is added between two convolutional layers and its weight is considered in max pooling layer. Therefore, interaction weight channel attention CNN is selected to extract semantic features in high level from sketches and 2D views.

Regularization coefficient is set to 0.04. Initial learning rate is set to 1E−4. The number of training iterations is set to 20. A total of 80% of training data are selected as training set and the rest is used as validation set. Figure 3 shows the loss of training set and validation set during the training process. Figure 4 shows accuracy of training set and validation set during the training process. It can be seen that the loss steadily decreases and convergence speed is fast under this learning rate. At the same time, accuracy achieves the best.

Alexnet algorithm uses CNN to extract features. CA algorithm adds channel attention into each convolutional layer to extract semantic features. ICA algorithm adds interactive attention module between the first convolutional layer and the second one to extract features. Hog algorithm extracts histograms of oriented gradients (*Lei et al., 2014*). LBP algorithm extracts local features (*Singh & Agrawal, 2016*). Gist algorithm uses gist feature. Shape distribution feature is only adopted in SD algorithm. Through sharing Vgg16 weights, siamese learning network is established. Joint feature mapping of sketches and views is realized. Based on joint feature distribution, the similarity is computed between sketch and 3D model to realize 3D model retrieval based on sketch. MFFR (multiple feature fusion retrieval) algorithm combines features extracted by ICA algorithm, Gist algorithm and SD algorithm.

**Peer**J Computer Science

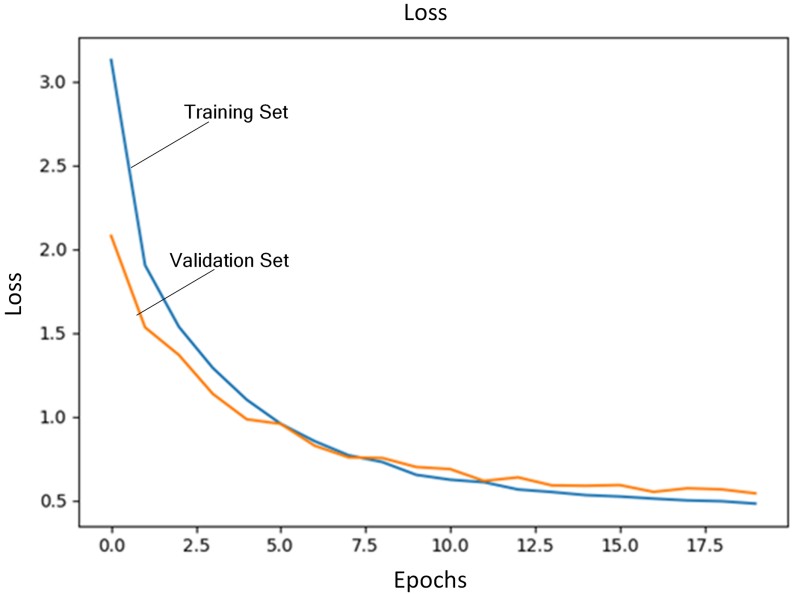

**Figure 3  Loss of training set and validation set.**

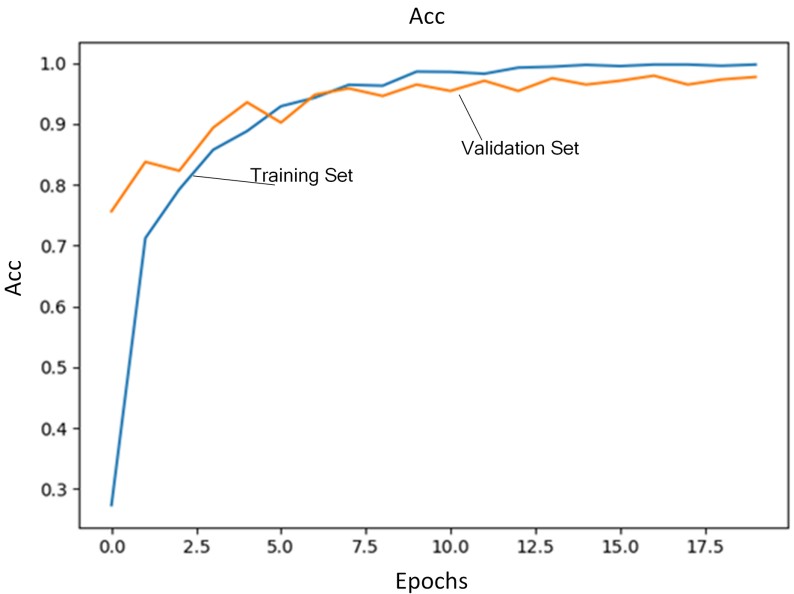

**Figure 4  Accuracy of training set and validation set.**

Gist feature and shape distribution features describe global information of sketch and view. Semantic feature denotes local details of sketch and view. These three features have their own advantages and disadvantages. We fuse semantic feature, gist feature and shape distribution features to describe sketch and 3D model adequately. The purpose is to make them complement each other for getting the best performance of 3D model retrieval. Discriminative ability of feature can be evaluated with NN, FT, ST, E(F) and DCG when it is applied to the retrieval process.

**Table 4 The performance of MFFR algorithm under different weights.**

| $w_1$ | $w_2$ | $w_3$ | NN | FT | ST | E(F) | DCG |
|-------|-------|-------|-------|-------|-------|-------|-------|
| 0.1 | 0.6 | 0.3 | 0.560 | 0.500 | 0.720 | 0.532 | 1.247 |
| 0.1 | 0.3 | 0.6 | 0.560 | 0.560 | 0.780 | 0.532 | 1.247 |
| 0.3 | 0.6 | 0.1 | 0.780 | 0.610 | 0.890 | 0.572 | 1.280 |
| 0.3 | 0.1 | 0.6 | 0.780 | 0.560 | 0.830 | 0.584 | 1.247 |
| 0.6 | 0.1 | 0.3 | 0.670 | 0.610 | 0.890 | 0.528 | 1.247 |
| 0.6 | 0.3 | 0.1 | 0.780 | 0.560 | 0.940 | 0.617 | 1.581 |

**Table 5 Evaluation of different algorithms on test data.**

| Method | NN | FT | ST | E(F) | DCG |
|--------|-------|-------|-------|-------|-------|
| Alexnet | 0.667 | 0.433 | 0.588 | 0.391 | 3.606 |
| CA | 0.667 | 0.411 | .0.633 | 0.444 | 2.863 |
| ICA | 0.556 | 0.577 | 0.766 | 0.533 | 3.250 |
| HOG | 0.333 | 0.320 | 0.530 | 0.271 | 2.464 |
| LBP | 0.111 | 0.120 | 0.240 | 0.111 | 0.387 |
| Gist | 0.444 | 0.330 | 0.533 | 0.338 | 1.452 |
| SD | 0.333 | 0.211 | 0.400 | 0.240 | 1.432 |
| Vgg16+Siamese | 0.778 | 0.531 | 0.401 | 0.452 | 3.565 |
| MFFR | 0.889 | 0.544 | 0.756 | 0.533 | 3.939 |

We need determine optimal values of $w_1$, $w_2$, $w_3$ in Formula (15) through experiments to make MFFR algorithm achieve the best. We set different values to parameters $w_1$, $w_2$, $w_3$, where the sum of $w_1$, $w_2$, $w_3$ is 1. Under different values of parameters $w_1$, $w_2$, $w_3$, MFFR algorithm is applied to search 3D models from test data which are the most similar with these nine models. NN, FT, ST, E(F) and DCG are used to evaluate the performance of MFFR algorithm as shown in Table 4.

From Table 4, MFFR algorithm all achieves the best at NN, ST, E(F) and DCG when $w_1 = 0.6$, $w_2 = 0.3$, $w_3 = 0.1$. FT of MFFR algorithm is 0.560, which is only 0.04 lower than optimal one. So, we set 0.6, 0.3, 0.1 respectively to $w_1$, $w_2$, $w_3$ in Formula (15). We can find that the weight of semantic feature is 0.6, the weight of gist feature is 0.3, and the weight of shape distribution feature is 0.1. The weight of semantic feature is the biggest. This shows that semantic feature has more influence on the performance of 3D model retrieval than gist feature and shape distribution feature.

MFFR algorithm is compared with Alexnet algorithm, CA algorithm, ICA algorithm, Gist algorithm, HOG algorithm, LBP algorithm, and SD algorithm, Vgg16+Siamese. Experimental results are shown in Table 5.

Experimental results show that when one retrieval result is returned, MFFR algorithm achieves the highest precision and precision is 0.889, followed by Vgg16+Siamese, Alexnet algorithm and CA algorithm. Precision of Vgg16+Siamese is 0.778. Precisions of Alexnet

and CA algorithm are all 0.667. ICA algorithm ranks the fourth, and precision is 0.556. When evaluation criteria are FT and ST, ICA algorithm has the highest recalls, which are 0.577 and 0.766 respectively. The second is MFFR algorithm, which are 0.544 and 0.756 respectively. When evaluation criteria is E(F), MFFR algorithm and ICA algorithm achieve the highest precision and precisions are both 0.533. HOG algorithm, LBP algorithm, Gist algorithm and SD algorithm have low performance. 3D model retrieval method based on neural network is better than traditional algorithms. ICA is better than CA at FT, ST and E (F). CA is only better than ICA at NN. When DCG is used as evaluation criteria, MFFR achieves the best and its DCG is 3.939, followed by Vgg16+Siamese, Alexnet algorithm and ICA algorithm. CA algorithm ranks the fourth. It shows that interactive attention model can focus on more useful information in process of extracting features and reduce the loss of information. The effect of feature extraction is good. However, the improved CNN still has certain limitations due to the lack of training data. The retrieval effect does not meet the expected standard when one result is only returned. MFFR combines semantic feature, shape distribution features and gist feature to retrieve 3D model, and its performance at NN criterion is improved. Compared with ICA, the performance of MFFR reduces a little under FT and ST criterion.

SD algorithm uses shape distribution features to describe sketch and views. Gist algorithm uses gist feature to describe sketch and view. Gist feature and shape distribution feature describe global information of sketch and view. Alexnet algorithm uses CNN to extract features from sketch and view. CA algorithm adds channel attention into each convolutional layer to extract semantic features from sketch and view. ICA algorithm adds interactive attention between the first convolutional layer and the second one to extract features from sketch and view. They can get local information of sketch and view. Vgg16 +Siamese uses siamese network to share the weights of two Vgg16. Joint feature mapping of sketches and views is built for retrieving 3D model based on sketch. From Table 5, we can find that the retrieval method based on local information achieves better than that based on global information. This is because that local information denotes details of sketch and view, which can distinguish sketch or view from others better. But, gist feature and shape distribution feature have global information. When global and local information are applied to 3D model retrieval, the performance is improved. So, MFFR achieves the best.

Figure 5 describes precisions of nine algorithms under different recall values.

In these nine algorithms, precision decreases gradually as recall increases. MFFR has the highest precision when recall is between 0 and 0.2. When recall is 0.1, Vgg16+Siamese has higher precision, which is better than Alexnet, ICA and CA. The reason is that Siamese network is used to share weights of two Vgg16 for building joint feature mapping of sketches and views better. SD algorithm is higher than HOG algorithm and LBP algorithm. This is because that shape distribution feature can give more accurate description of sketch and view than HOG and LBP features. When recall is 0.2, ICA is better than Alexnet and CA. When recall is between 0.3 and 1, ICA has the highest precision and is followed by MFFR. SD algorithm, HOG algorithm, LBP algorithm and Gist algorithm have lower precision. Experimental results show that deep learning algorithms are better than

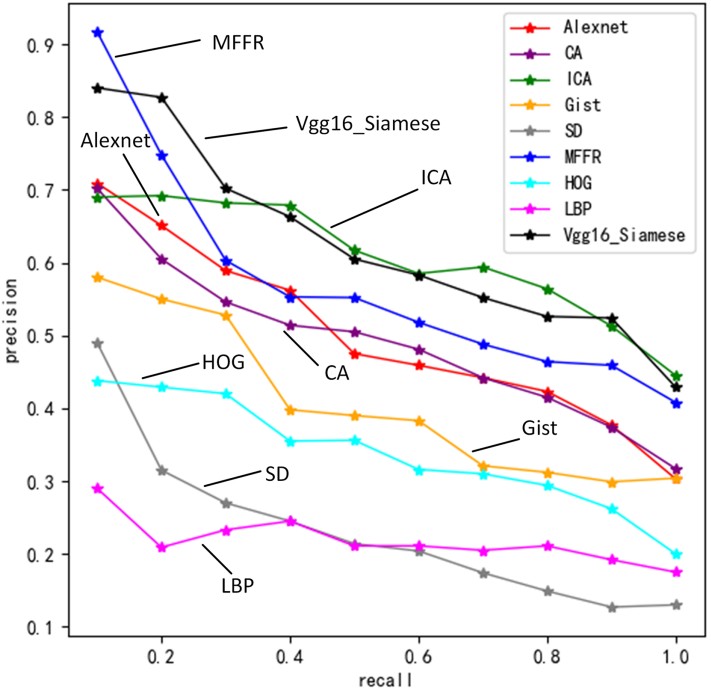

**Figure 5 Retrieval precision of nine algorithms under different recall rates.**

traditional ones at precision. When semantic feature, shape distribution features and gist feature are fused, information of sketch and view can be described accurately.

Figure 6 shows the performance of these nine algorithms at E(F) for four categories of 3D models. It can be seen that the performance of these nine algorithms is significantly different under different categories. For some categories such as chair, ICA is better than other algorithms. For some categories such as person, the performance of ICA does not reach the expected results. But, Gist algorithm has the best performance. For some complex categories, neural network is not trained sufficiently and does not perform well in process of extracting features. Traditional method of feature extraction does not require training process and has relatively good effect. Therefore, it is further proved that MFFR algorithm can improve the performance of 3D model to a certain extent.

It can be found that Alexnet algorithm, CA algorithm, ICA algorithm, Vgg16+Siamese and MFFR algorithm achieve better than Gist algorithm and SD algorithm. This is because that deep learning algorithm has a good effect in process of extracting 3D model's features. It can be found that ICA algorithm and MFFR algorithm achieve better than Alexnet algorithm and CA algorithm. This is because that attention mechanism is added into CNN and uses adaptive weight to reduce the loss of information. It greatly improves the performance of retrieving some kinds of models. MFFR algorithm fuses multiple features and the performance of 3D model is improved at criterion NN.

3D model can be denoted as its 2D views. The number of 2D views has influence on the expression of 3D model. More views can describe details of 3D model. But, it will introduce noises into the process of training the proposed network. At the same time, time and space

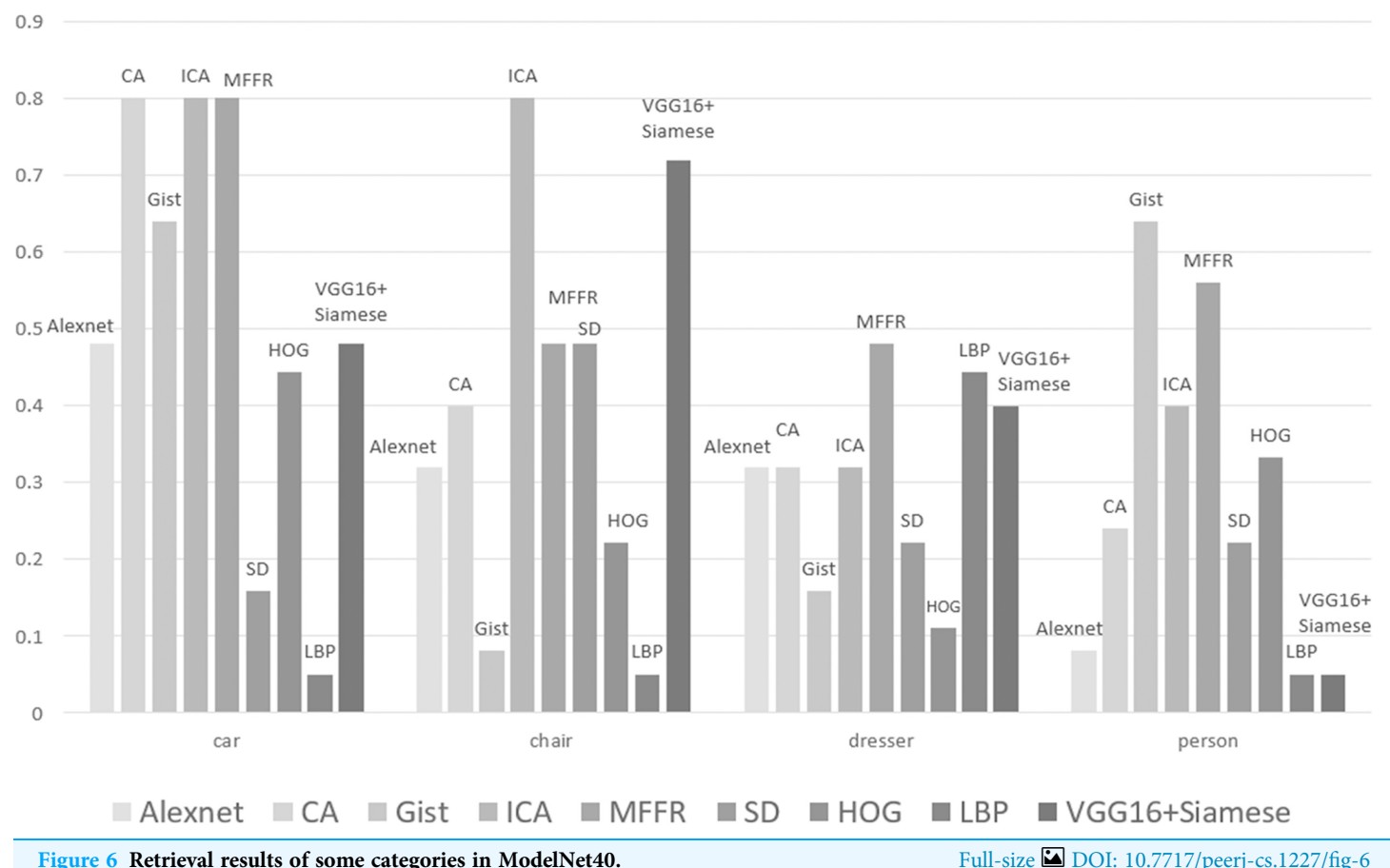

**Figure 6 Retrieval results of some categories in ModelNet40.**

**Table 6 Accuracy of the proposed network under different view number.**

| View number | 3 | 4 | 6 | 8 | 9 | 12 | 15 | 18 |
|---|---|---|---|---|---|---|---|---|
| Accuracy | 0.8892 | 0.8864 | 0.9832 | 0.9774 | 0.9851 | 0.9882 | 0.9905 | 0.9910 |

complexity will increase greatly. 450 3D models are used to testify accuracy of the proposed network under different view number. We set 3, 4, 6, 8, 9, 12, 15, 18 to view number respectively. 2D views of these 450 3D models are extracted under different view number. Then, they are used to train and test the proposed network. Accuracies of the proposed network under different view number are shown in Table 6.

From Table 6, it can be seen that accuracy of the proposed network increases with the increase of view number. When view number is 6, the classification accuracy achieves 0.9832. When we use 18 views to express 3D model, accuracy is 0.9910. accuracy is only increased by 0.79%, but the number of views increases three times. The complexity of training the network will increase greatly. When four views and eight views are adopted to express 3D model, accuracy is 0.8864 and 0.9774 respectively. So, we use six views to express 3D model in this article.

**Table 7 Evaluation of MFFR algorithm on different test dataset.**

| Dataset | NN | FT | ST | E(F) | DCG |
|---|---|---|---|---|---|
| OpenSketch | 0.667 | 0.160 | 0.132 | 0.134 | 3.614 |
| CADNET | 1 | 0.951 | 0.889 | 0.635 | 4.255 |
| MCB | 0.727 | 0.331 | 0.248 | 0.395 | 2.966 |

OpenSketch dataset (*Gryaditskaya et al., 2019*) is divided into training data and test data. There are nine categories including bumps, flange, hairdryer, house, mixer, mouse, potato_chip, shampoo_bottle, tubes, vacuum_cleaner, waffle_iron, wobble_surface. Training data is used to optimize MFFR algorithm. Test data is adopted to testify its performance and experimental results are shown in Table 7. A total of 11 categories are selected from MCB dataset (*Kim et al., 2020*) including Bushes, Clamps, Fan, Lever, Lockwashers, Pulleys, Split_pins, Springs, Switch, Threaded_rods, Wheel. Training data is used to optimize MFFR algorithm. Test data is adopted to testify its performance and experimental results are shown in Table 7. Nine categories are selected from CADNET dataset (*Manda, Bhaskare & Muthuganapathy, 2021*) including Bearing_Blocks, Contact_Switches, Curved_Housings, Gear_like_Parts, Handles, Intersecting_Pipes, Oil_Pans, Posts, Rocker_Arms. Training data is used to optimize MFFR algorithm. Test data is adopted to testify its performance and experimental results are shown in Table 7.

From Table 7, we can find that the performance of MFFR algorithm on OpenSketch dataset, CADNET dataset, MCB dataset is all good at NN criterion. MFFR algorithm achieves the best on CADNET dataset at NN, FT, ST, E(F) and DCG.

# CONCLUSIONS

This article proposes a method of 3D model retrieval based on interactive attention CNN and multiple features. 3D model is projected into a set of 2D views, and edge detection algorithm is used to extract line graph of 2D view. CNN with interactive attention is used to extract semantic features from sketches and 2D views. 3D shape distribution descriptors are improved to be suitable for 2D views. Then, shape distribution features and gist feature are extracted from 2D views. Semantic features, shape distribution features and gist feature are merged to retrieve 3D model similar to sketch. Compared with the retrieval methods based on single features, the proposed one can improve the performance of 3D model retrieval.

The 3D model is rotated one circle to extract a set of 2D views. We represent a 3D model with its 2D views. In fact, contours of 2D views have important influence on shape of 3D model. Internal part of 2D view has little influence on shape expression of 3D model. So, we select 2D views' contours to train the proposed network. At the same time, contours are combined with gist feature and shape distribution features to express 3D model further.

The novelty of the proposed method is that interactive attention CNN is adopted to extract features from 2D views of 3D model. Gist algorithm and SD algorithm are used to extract global features. Euclidean distance is adopted to calculate the similarity of semantic

feature, the similarity of gist feature and the similarity of shape distribution features between sketch and 2D view. Then, the weighted sum of three similarities is used to compute the similarity between sketch and 2D views for retrieving 3D model. It can combine global and local information for retrieving 3D model.

## ACKNOWLEDGEMENTS

I would like to thank Professor Zhang for his works on the article and Master Jia for conducting experiments.

### Funding

This work was supported by Heilongjiang Provincial Natural Science Foundation of China (No. LH2022F030). The funders had no role in study design, data collection and analysis, decision to publish, or preparation of the manuscript.

### Grant Disclosures

The following grant information was disclosed by the authors:
Heilongjiang Provincial Natural Science Foundation of China: LH2022F030.

### Competing Interests

The authors declare that they have no competing interests.

### Author Contributions

- Xue-Yao Gao conceived and designed the experiments, analyzed the data, performed the computation work, authored or reviewed drafts of the article, proposed the idea that interactive attention CNN and weighted similarity calculation are combined for retrieving 3D models, and approved the final draft.
- Wenhui Jia performed the experiments, analyzed the data, performed the computation work, prepared figures and/or tables, and approved the final draft.
- Chun-Xiang Zhang conceived and designed the experiments, performed the experiments, analyzed the data, performed the computation work, prepared figures and/or tables, authored or reviewed drafts of the article, and approved the final draft.

### Data Availability

   ModelNet40 is available at Princeton ModelNet: http://modelnet.cs.princeton.edu/.
   Shrec13 is available at SHREC 2013 Track Proposal: Large-Scale Partial Shape Retrieval Track Using Simulated Range Images: http://dataset.dcc.uchile.cl/.

### Supplemental Information

Supplemental information for this article can be found online at http://dx.doi.org/10.7717/peerj-cs.1227#supplemental-information.

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
