# Peer review of "D model retrieval based on interactive attention CNN and multiple features"

_PeerJ Computer Science, doi:10.7717/peerj-cs.1227_

## Round 0.1 · original submission · Major Revisions

The reviewers raised a number of concerns for the paper but they also find the paper has some merits. I encourage you to address all these concerns and provide a revised version. Thanks.

Reviewer 1 ·

Basic reporting

The language in which the paper is written is clear, unambiguous and professional. However, some minor mistakes were noted throughout the paper that can be easily rectified.
Concerning the section pertaining to related work in the introduction, it is sufficient. Nonetheless, it can be improved by citing other related works.
The overall structure of the paper conforms to the PeerJ standards that are cited in the editorial criteria.
All of the figures are relevant, of high quality and perfectly described and labelled, except for figure 8 is a replication of figure 7. Figure 8 “should describe retrieval precision of 6 algorithms under different recall rates” as it is pointed out in the paper, not be a replica of figure 7.
The authors used some acronyms throughout the paper without initially specifying what they stand for (e.g. MFFR).

Experimental design

The original primary research of the paper is within the scope of this journal.
The paper addresses the issue of 3D object retrieval based on sketch. To retrieve 3D objects similar to a given sketch, the authors represent each 3D object by a set of views, and then they combine semantic features, shape distribution one and GIST one to characterize the 3D objects’ views and the sketches, so that the shape-matching problem between sketches and 3D objects is transformed into similarity measuring between sketches and 3D objects’ views. However, the authors did not clearly justify the choice of this combination of the features. They mentioned that “Feature of good discriminative ability is assigned with high weight. Otherwise, it is assigned with low weight”, without specifying how the discriminative ability can be measured. In equation 21, the similarity weighting parameter is not detailed. The authors should specify how this parameter can be chosen within an equation and how it can be calculated.
More tests should be conducted in order to choose the most suitable hyperparameter of the used CNN.
Despite the abovementioned remarks, the paper’s question is well defined and clearly stated. Moreover, the research fills an identified knowledge gap.

Validity of the findings

The authors utilized objects from SHREC13 and ModelNet40 to evaluate the proposed approach. However, there is no mention of the precise number of the used objects and sketches. The description of the used objects lacks precision and meticulousness; it should include the exact number of used objects in each category. Selecting objects from both data sets makes it challenging for other researchers to compare the results of this paper with theirs.
The authors used shape distribution algorithm and a GIST one (plus another one) as the basis of their proposed method. However, in the discussion, the authors concluded that the two aforementioned algorithms used in their approach perform poorly compared to “Alexnet algorithm, CA algorithm, ICA algorithm”, according to the tests they conducted.

Reviewer 2 ·

Basic reporting

1. The Title and Abstract of the paper do not seem to clearly reflect the ideas conveyed in the paper. I request the authors to come up with a better title and rephrase the abstract accordingly.

2. The paper could have been well written - both in terms of the English language and also in terms of
presentation. The paper needs to be thoroughly edited and proofread. I would recommend that the authors take help from a native English speaker or use a third party service. Many typos
still exist in the paper. To mention a few (in the Abstract),
- "To solve this problem, we combine semantic feature, shape distribution one and gist one to..." - Please rephrase the statement, since it is hard to understand what is being conveyed
- Abbreviate the notations in first use (eg. CNN, NN, FT, ST and F)
- The authors simply mention that "3D models are widely applied in our daily life now." Where are they applied? It would be better if the authors provided some examples

3. The Literature Review is not comprehensive. It should be updated to the current state-of-the-art. Some suggested references are
- NeuralContours [Liu, Difan, et al. "Neural contours: Learning to draw lines from 3d shapes." Proceedings of the IEEE/CVF Conference on Computer Vision and Pattern Recognition. 2020.]
- Qin, Fw., Gao, Sm., Yang, Xl. et al. A sketch-based semantic retrieval approach for 3D CAD models. Appl. Math. J. Chin. Univ. 32, 27–52 (2017). https://doi.org/10.1007/s11766-017-3450-3
- Li, Bo, et al. "A comparison of methods for sketch-based 3D shape retrieval." Computer Vision and Image Understanding 119 (2014): 57-80.
- Tangelder, Johan WH, and Remco C. Veltkamp. "A survey of content based 3D shape retrieval methods." Multimedia tools and applications 39.3 (2008): 441-471.

4. The figures need to be improved. Figure 5 has poor resolution. In Figure 9, the legend covers
the bar plot and hence the actual graph cannot be seen.

Experimental design

1. The article lies within the Aims and Scope of the Journal

2. The method used by the authors is explained quite well

3. The authors have tried to address a very interesting and relevant research problem. There is a lot of potential application to this research problem. However, the results presented in the Results section (Table 3) for comparison are quite old. The current field of research is constantly updated every day with newer ideas and technologies. The authors should update the paper to provide a more rigorous comparison of their results with the latest methods.

4. The datasets used in this paper like SHREC13 and ModelNet40 are standard datasets. But the SHape REtrieval Contests (SHREC) conducted in the year 2014 improves upon the SHREC 2013 track by including Extended Large Scale Sketch-Based 3D Shape Retrieval track, with a larger number of data samples. However, the authors only use the data from SHREC 2013 track. What is the reason for this? Also, other popular datasets such as
- OpenSketch [Gryaditskaya, Yulia, et al. "OpenSketch: a richly-annotated dataset of
product design sketches." ACM Trans. Graph. 38.6 (2019): 232-1.]
- ProSketch [Zhong, Yue, et al. "Towards Practical Sketch-based 3D Shape Generation:
The Role of Professional Sketches." IEEE Transactions on Circuits and Systems for Video
Technology (2020)]
should also be made use of and the results should be tested on these datasets as well.

5. This is with respect to the number of views of a 3D model. You mention that 6 view images are obtained by rotating the 3D model and taking a view for every 60 degrees. How did you justify that 6 view images are sufficient? Did you experiment with the number of views? Does using a more number of view images create redundancy? The paper on MVCNN [Su, Hang, et al. "Multi-view convolutional neural networks for 3d shape recognition." Proceedings of the IEEE international conference on computer vision. 2015] uses 12 views per 3D model. The paper on Light Field Descriptor [Chen, Ding-Yun, et al. "On visual similarity based 3D model retrieval." Computer graphics forum. Vol. 22. No. 3. Oxford, UK: Blackwell Publishing, Inc, 2003.] uses 20 views per 3D model. So how is your idea of using just 6 views different from these methods, and how do you justify your choice?

6. There is some ambiguity within the paper upon how many 2D views are used. The authors indicate that 6 views are used per 3D model in Section 3. However, Figure 2 use the singular word '2D view' and not '2D views'. Does this mean that each view is being processed separately? Please resolve this inconsistency.

Validity of the findings

1. The authors claim in the title that they propose 'A Novel Method of 3D Model Retrieval ...'. But the proposed CNN architecture is pretty staright forward and simple. Many architectural novelties exist such as the use of Residual Connections in [He, Kaiming, et al. "Deep residual learning for image recognition." Proceedings of the IEEE conference on computer vision and pattern recognition. 2016.], or the use of Inception module in [Szegedy, Christian, et al. "Going deeper with convolutions." Proceedings of the IEEE conference on computer vision and pattern recognition. 2015.] etc. I find it hard to see any novelty proposed in the paper - either from the neural network architecture standpoint nor from the dataset standpoint. The other ideas used such as similarity metrics are also standard methods in the literature.

2. The validity of the proposed method is not clear when it comes to 3D models of Engineering shapes and components. The authors should explain if the developed approach work on all kinds of 3D
inputs. The authors only seem to test their method on the ModelNet40 dataset, which are in the
point cloud format. No mention is made of the recent 3D CAD model datasets such as
- ABC Dataset (Koch, Sebastian, et al. "Abc: A big cad model dataset for geometric deep
learning." Proceedings of the IEEE/CVF Conference on Computer Vision and Pattern
Recognition. 2019.)
- MCB dataset (Kim, Sangpil, et al. "A large-scale annotated mechanical components benchmark
for classification and retrieval tasks with deep neural networks." Proceedings of 16th European
Conference on Computer Vision (ECCV). 2020.)
- CADNET dataset (Manda, Bharadwaj, Pranjal Bhaskare, and Ramanathan Muthuganapathy.
"A Convolutional Neural Network Approach to the Classification of Engineering Models." IEEE
Access 9 (2021): 22711-22723.)

Additional comments

1. The paper is unnecessarily lengthy in many places. For example, in the Methods Section, the details about backpropagation, what is a convolution layer and what is a pooling layer etc are not required. The usage of CNNs has been going on for many years now and these terms are pretty standard to researchers. There is no need to describe these details at length.

Reviewer 3 ·

Basic reporting

(1) The literature survey is not enough. The problems of current sketch-based 3D model retrieval are not analyzed deeply enough.
(2) The written English of this paper should be improved. There are amounts of syntax errors, which make it difficult to understand this paper.
(3) The equation can be improved.
(4) The fonts are inconsistent and some letters do not set to subscripts in the figure 3.

Experimental design

Why 3D models are represented with six 2D views, how about four or eight 2D views? You should compare them in experiments.

Validity of the findings

(1) 2D views are represented with contours, which loses useful information. How to solve this problem?
(2) Semantic features, GIST features and shape distribution features are extracted from 3D models, but these feature extraction methods are all existing methods. I cannot see any novelty.

---

## Round 0.2 · Major Revisions

I have taken into all reviewers' comments.I encourage the authors to address all reviewers' comments in the revised version.

Reviewer 1 ·

Basic reporting

- The language in which the paper is written is clear, unambiguous, and professional.
- Concerning the section pertaining to related work in the introduction, it is sufficient.
- The overall structure of the paper conforms to the PeerJ standards that are cited in the editorial criteria.
- All of the figures are relevant, of high quality, and perfectly described and labelled.

Experimental design

- The original primary research of the paper is within the scope of this journal.
- The paper’s question is well-defined and clearly stated. Moreover, the research fills an identified knowledge gap.

Validity of the findings

The manuscript has been well improved, and I think most of my comments have been addressed with either new analysis or necessary discussions.

Reviewer 2 ·

Basic reporting

The writing is a lot better compared to the previous version. The ambiguities that existed before have been clarified now.

The literature survey has also been updated as per the comments of the reviewers. The figures and the results tables were also updated in a few places.

Overall, I am satisfied with the Basic Reporting.

Experimental design

No comment.

Validity of the findings

I am satisfied with the changes made to the paper. The experiments section has been revamped as per the reviewer comments.

Additional comments

The paper looks in a much better shape as compared to the previous version. I really appreciate the authors' efforts in making the changes as per the comments of the reviewers.

I am happy to recommend the paper for acceptance, provided the following changes are made:

(1) The introduction section looks lengthy. I would suggest the authors to split it into two sections - Introduction and Literature Review - for the ease of readability

(2) The novelty of the proposed approach is mentioned in the conclusion section, but is not clearly mentioned in the Introduction. I recommend the authors to add these details in the Introduction that gives the key highlights of the paper and its contributions in a numbered list. This will benefit the manuscript.

Reviewer 3 ·

Basic reporting

(1) This paper needs to be thoroughly modified. I mentioned in the last version that there were many grammatical errors in this paper. In this new version, there are still many grammatical errors.
(2) The figures still need to be improved. In Figure 1, the initials of words are capitalized except for ‘attention’, and brackets should not be italicized, and letters should not be covered by arrows. The Figure 2 is not clear, and all initials of words should be capitalized.

Experimental design

(1)There are many sketch-based 3D model retrieval methods with good performance, such as JFM and DCSSE. But the method proposed in this paper is not compared with these methods, and do not use the evaluation criteria E and DCG.

Validity of the findings

(1) The method proposed in this paper is lack of innovations. The authors said that the innovation of this paper is the interactive attention CNN. In your paper, you said the interactive attention module is introduced into the framework as shown in Figure 1. But I cannot find the interactive attention module in Figure 1. Is it the attention part? This paper does not clearly explain how the interactive attention module works.

---

## Round 0.3 · accepted · Accept

Reviewers' comments have been addressed. The paper can be accepted.

Reviewer 2 ·

Basic reporting

NA

Experimental design

NA

Validity of the findings

NA

Additional comments

I am satisfied with all changes made to the manuscript. I am happy to recommend the paper for accpetance.